

# Expression, refolding and spectroscopic characterization of fibronectin type III (FnIII)-homology domains derived from human fibronectin leucine rich transmembrane protein (FLRT)-1, -2, and -3

Lila Yang[1,*], Maria Hansen Falkesgaard[2,*], Peter Waaben Thulstrup[1], Peter Schledermann Walmod[2], Leila Lo Leggio[1] and Kim Krighaar Rasmussen[1,2]

[1] Biological Chemistry, Department of Chemistry, University of Copenhagen, Copenhagen, Denmark
[2] Laboratory of Neural Plasticity, Department of Neuroscience and Pharmacology, Faculty of Health and Medical Sciences, University of Copenhagen, Copenhagen, Denmark
[*] These authors contributed equally to this work.

Corresponding authors
Leila Lo Leggio, leila@chem.ku.dk
Kim Krighaar Rasmussen, kkr@chem.ku.dk

## ABSTRACT

The fibronectin leucine rich transmembrane (FLRT) protein family consists in humans of 3 proteins, FLRT1, -2, and -3. The FLRT proteins contain two extracellular domains separated by an unstructured linker. The most membrane distal part is a leucine rich repeat (LRR) domain responsible for both *cis-* and *trans-*interactions, whereas the membrane proximal part is a fibronectin type III (FnIII) domain responsible for a *cis-*interaction with members of the fibroblast growth factor receptor 1 (FGFR1) family, which results in FGFR tyrosine kinase activation. Whereas the structures of FLRT LRR domains from various species have been determined, the expression and purification of recombinant FLRT FnIII domains, important steps for further structural and functional characterizations of the proteins, have not yet been described. Here we present a protocol for expressing recombinant FLRT-FnIII domains in inclusion bodies in *Escherichia coli*. His-tags permitted affinity purification of the domains, which subsequently were refolded on a Ni-NTA agarose column by reducing the concentration of urea. The refolding was confirmed by circular dichroism (CD) and $^1$H-NMR. By thermal unfolding experiments we show that a strand-strand cystine bridge has significant effect on the stability of the FLRT FnIII fold. We further show by Surface Plasmon Resonance that all three FnIII domains bind to FGFR1, and roughly estimate a $K_d$ for each domain, all $K_d$s being in the $\mu$M range.

## INTRODUCTION

The fibronectin leucine rich transmembrane (FLRT) protein family was first described in 1999. In mammals it includes three proteins, FLRT1, -2, and -3 (*Lacy et al., 1999*) each comprised of an extracellular, membrane-distal leucine-rich repeat (LRR) domain (*Seiradake et al., 2014*) followed by a fibronectin type III (FnIII) domain, a transmembrane region and a cytoplasmic tail without obvious sequence motifs (*Karaulanov, Böttcher & Niehrs, 2006*).

A number of extracellular binding partners to FLRT proteins have been reported, including Myelin-associated glycoprotein, Brother of CDO, netrin receptors, and the synaptic, G-protein-coupled receptors, latrophilins (reviewed by *Winther & Walmod, 2014*). Another group of proteins that interacts with FLRTs is the Fibroblast Growth Factor Receptor (FGFR) family. FGFRs constitute a family of trans-membrane receptors belonging to the receptor tyrosine kinase superfamily. They consist of an extracellular domain comprising three immunoglobulin (Ig) domains, a transmembrane domain (TMD), and an intracellular tail including a split tyrosine kinase domain. Ig domain 2 and 3 and the linker between them are known to be involved in binding to the classical FGFR ligands, FGFs, whereas Ig domain 1 has an autoinhibitory function on ligand binding (*Gong, 2014*). Following receptor dimerization, the intracellular kinase domains are trans-activated. The activated receptors in turn initiate downstream signaling through various pathways to generate diverse cellular responses, including cell growth, differentiation, migration and cell survival (*Slack et al., 1996*; *Martin, 1998*; *Friesel & Maciag, 1999*; *Groth & Lardelli, 2002*). Consequently, FGFR are implicated in a number of diseases including cancer, schizophrenia and Parkinson's disease (*Jin et al., 2005*; *Krejci et al., 2009*; *Turner & Grose, 2010*; *Haugsten et al., 2010*; *Van Scheltinga, Bakker & Kahn, 2010*). In addition to their interaction with FGFs, the extracellular domains of FGFRs can bind several proteins located in the plasma membrane, including a number of cell adhesion molecules among which FLRTs.

The FnIII domain of *Xenopus* FLRT3 (XFLRT3) is reported to bind directly to FGFR1 (*Böttcher et al., 2004*), and a later study has demonstrated that all three mouse FLRTs act as regulators of FGFR-mediated signaling through direct interaction with FGFR1 independently of prior FGFR activation (*Haines et al., 2006*). Surprisingly, a more recent study implicates both the LRR and the cytoplasmic tail of mouse FLRT2 in a direct interaction with FGFR2 (*Wei et al., 2011*).

FLRT-FGFR-mediated signaling is important e.g., for craniofacial development (*Wei et al., 2011*) and the folding of the cerebral cortex (*Del Toro et al., 2017*). The three mammalian FLRTs have different but occasionally overlapping expression patterns. Since they all can bind e.g., FGFR1 any specificities in the signaling mediated by FLRT-FGFR interactions would therefore require differences e.g., in the affinities between FLRTs and FGFRs. FLRT-FGFR interactions, and in particular the interactions between FGFRs and FLRT FnIII domains, need therefore to be studied in more detail.

The LRR regions of FLRT proteins have been produced recombinantly and crystallized in complex with netrin receptors (*Seiradake et al., 2014*) and latrophilins (*Jackson et al., 2015*). In contrast, the expression and purification of FLRT FnIII domains *in vitro* have

been so far not been reported and the detailed interaction between FGFR and FLRTs remains to be elucidated.

Here we present a procedure for the expression and purification of FLRT FnIII domains. One aim of the study was to eventually produce deuterated proteins suitable for small angle neutron scattering, and hence *Escherichia coli* was chosen as an expression system.

## MATERIALS AND METHODS

### Construction and expression of FLRT-FnIII domains

Genes coding for human FLRT1, -2, and -3 were purchased from Source BioScience GmbH (Germany). Using polymerase chain reaction (PCR) with oligonucleotides primer (FLRT1-FnIII forward: 5′-GATGGCGCCAAGACCCTGGCC-3′, reverse: 5′-GGTA GGGCCATAGCTGTCGGC-3′, FLRT2 FnIII forward: 5′-CCTATTTCTGAACGGATCC AGC-3′, reverse: 5′-CAGATAGGAGGCATGGGTGGT-3′, FLRT3-FnIII: 5′-AGTCCCTCA AGAAAAACAATTAC-3′, reverse: 5′-CATTCGAAGGGGTGCAGTTTCAGT-3′) cDNA for all three FLRT-FnIII domains was amplified and cloned into pET-DEST42 Gateway (Invitrogen) destination vector containing ampicillin resistance.

### Protein expression

All the constructs were transformed into *E. coli* BL21 (DE3) (Novagen) or Rosetta (DE3) (Novagen) and selected with appropriated antibiotics. A total of 10 ml starter culture of Luria Bertani (LB) broth containing 100 µg/ml ampicillin or 50 µg/ml kanamycin was inoculated with a single colony, and grown at 37 °C overnight with 250 rpm shaking.

Afterwards, 1 L LB containing 100 µg/ml ampicillin or 50 µg/ml kanamycin was inoculated with the overnight starter culture. The culture grew at 37 °C with 250 rpm shaking until an $OD_{600}$ of 0.6 was reached. Overexpression was induced by adding isopropyl β-D-1-thiogalactopyranoside (IPTG) to a final concentration of 1 mM, and the cultures were allowed to overexpress for 20 h at 15 or 25 °C. The cells were harvested by centrifugation at $6,000 \times g$ for 10 min. Cell pellets were frozen for minimum 15 min at −20 °C. Thawed cell pellets were resuspended in denaturing buffer (8M urea, phosphate buffered saline (PBS), pH 7.4, 1 mM Dithiothretol (DTT)) at 5 ml/g wet weight, and sonicated for 7 min with 10 s pulse and 10 s pause with an amplitude set to 50%. Following sonication the cell paste was stirred for 60 min at room temperature to solubilize inclusion bodies. The lysate containing soluble inclusion bodies was centrifuged at $20,000 \times g$ for 30 min at +4 °C to pellet insoluble material. The supernatant was collected and filtered with 0.45 µm sterile filter before purification. 20% SDS-PAGE was used to follow purification and verify protein purity with typical sample loading of 0.5–1 mg/ml protein. SDS-PAGE gels were stained overnight with InstantBlue[TM] (Expedeon).

### Purification and refolding of FLRT FnIII domains

FnIII domains from filtrates were immobilized on a 1 ml HP HisTrap Ni-NTA column (1 for each protein) (GE Healthcare, Little Chalfont, UK) coupled to an Äkta purify system. The column was equilibrated with 10–15 column volumes of denaturing buffer (8 M urea, PBS, 1 mM DTT, pH 7.4) with flow rate 1 ml/min. Following equilibration the filtrate was

loaded onto the column, and washed with wash buffer (8 M urea, PBS, 1 mM DTT, 20 mM imidazole, pH 7.4) for at least 30 column volumes until the $A_{280}$ nm and $A_{260}$ nm were stabilized. Subsequently, the washed proteins were refolded on the column by applying PBS (pH 7.4), thus generating a decreasing gradient of urea and DTT (from 8 M urea and 1 mM DTT to 0 M in 30 min). A second wash step was included with wash buffer (5 × PBS and 20 mM imidazole pH 7.4). Before eluting the protein, PBS was run through the column to decrease the salt concentration. The protein was eluted with elution buffer (PBS pH 7.4 containing 500 mM imidazole) for 5 column volumes. The purification was monitored using ultraviolet absorption at 280 nm and 260 nm. The refolded proteins were further purified by size-exclusion chromatography (SEC) using a 320 ml or 120 ml Superdex 75 16/60 GL column equilibrated with regular PBS. The collection was monitored at $A_{280}$ nm and $A_{260}$ nm. All refolding and purification procedures were carried out at 4 °C and the purified proteins were stored at −80 °C until use.

## Characterization of FLRT FnIII domains
### Matrix-assisted laser desorption/ionization time-of-flight (MALDI-TOF)
The masses of the FLRT domains were determined by MALDI-TOF-MS. 1 μl sample and 1 μl a-Cyano-4-hydroxycinnamic acid (HCCA) matrix were mixed and dispensed onto a MALDI target and allowed to dry out at room temperature. Protein masses were acquired in linear mid-mass positive mode, averaging 2,000 laser shots per MALDI-TOF spectrum. Calibration mixtures (Sigma-Aldrich, St. Louis, MO, USA) were used to calibrate the spectrum to a mass tolerance within 2 Da. The mass range was 5,000–14,000 Da.

### Circular dichroism spectroscopy
The Far-UV CD measurements were performed using a Jasco J-815 spectropolarimeter at 20 °C in a 1 mm path length quartz cuvette from Hellma over the wavelength range 190–260 nm. A resolution of 1 nm and a scan speed of 20 nm/min were employed. At least three consecutive scans were obtained, the resulting data was smoothed and the contribution of the buffer blank was subtracted. The monitored scans were averaged. Spectra were analyzed for secondary structural distribution using the Beta Structure Selection, BeStSel, server (*Micsonai et al., 2015*). Variable temperature measurements were carried out in the absence or presence DTT within the temperature ranges of 20–90 and 90−20 °C, at fixed wavelengths of 213 nm and 228.5 nm. Monitoring of CD at both wavelengths gave comparable results, but as the CD signal at 228.5 nm had a higher magnitude these data are presented. The temperature-dependent CD signal was normalized to vary between 1 and 0 for the initial and final data in the series for each protein sample, representing a fully folded and fully unfolded state. The (linear) temperature dependency of folded or unfolded protein was not taken into account.

### 1D-NMR
For 1D-NMR measurements, 50 μl $D_2O$ (Invitrogen) and 500 μl of the concentrated sample of protein were transferred to an NMR tube. Following the preparation of NMR samples a 1D-spectrum was recorded at 298 K on a Varian 750 MHz spectrometer using a cold probe.

### Template-based protein structure prediction

Primary sequences of all 3 FnIII domains (FLRT1; [405]GDGAKTLAIHVKALTADSIRITWKA
TLPASSFRLSWLRLGHSPAVGSITETLVQGDKTEYLLTALEPKSTYIICMVTMETSNAYVA
DETPVCAKAETADSYGPT[508], FLRT2; [420]GPISERIQLSIHFVNDTSIQVSWLSLFTVMAYK
LTWVKMGHSLVGGIVQERIVSGEKQHLSLVNLEPRSTYRICLVPLDAFNYRAVEDTICSE
ATTHASYL[519] and FLRT3; [404]GSPSRKTITITVKSVTSDTIHISWKLALPMTALRLSWLKL
GHSPAFGSITETIVTGERSEYLVTALEPDSPYKVCMVPMETSNLYLFDETPVCIETETAPL
RM[506]) were input to the I-TASSER server (*Yang et al., 2014*) for template-based modeling.

### Interactions studies with Fibroblast Growth Factor Receptor1 (FGFR1)

Binding studies were performed using a BIAcore 2000 instrument (BIAcore) at 25 °C
using 10-mM PBS (pH 7.4) as running buffer. A CM4 sensor chip was immobilized with
approximately 1300 RU of FGFR1β (IIIc)/Fc chimera (R&D Systems, Minneapolis, MN,
USA) and 750 RU recombinant human IgG$_1$ Fc, Cat. No. 110-HG (carrier-free; R&D
Systems, Minneapolis, MN, USA). The FGFR1 and IgG$_1$ proteins were immobilized by an
amine coupling kit (BIAcore) with flow rate 5 μl/min. FLRT1, -2 and -3 FnIII domains
in PBS (pH 7.4) were injected at 20–30 μl/min. The binding curves were analyzed by
BIAevaluation version 4.1 software. A steady state affinity analysis was used to estimate $K_d$.

## RESULT AND DISCUSSION

### Expression of FLRT FnIII domains

Purified pET-DEST42 destination vectors containing each relevant non-codon optimized
*FLRT FnIII* gene were transformed into *E.coli* strain BL21 (DE3) or the rare codon
harboring Rosetta (DE3), which were then used as expression hosts of recombinant FnIII
domains for all three FLRT domains. As standard conditions for protein expression
proteins, the strains harboring the plasmids were cultured at 37 °C until an OD$_{600}$ of 0.6
was reached. At this density the cultures were induced with IPTG, and temperature was
adjusted for expression. However, no obvious overexpression could be visualized for any
of the constructs. Although several expression parameters (different temperatures, *E. coli*
bacterial strains, and OD$_{600}$ values before IPTG induction) were tested, the non-codon
optimized genes cloned in to the pET-DEST42 vector gave no (FLRT2 FnIII) or little
(FLRT1 FnIII and FRLT3 FnIII) protein expression. Furthermore, the various changes in
expression conditions did not result in soluble protein, and the FnIII domains were always
expressed as inclusion bodies.

Since attempts to express the recombinant proteins from non-codon optimized
constructs had limited success, codon optimized genes were designed and purchased
from GenScript (Piscataway, NJ, USA). The genes were synthesized and cloned into the
pET-30-a(+), a vector that like pET-DEST42 contains the bacteriophage T7 polymerase
promoter (*Studier & Moffatt, 1986*). However, pET-30-a(+) and pET-DEST42 contain
different antibiotic resistance (kanamycin and ampicillin, respectively).

In contrast to the former expression experiments with the non-codon optimized
constructs, the codon optimized constructs were expressed with higher yields. The proteins
were still expressed as inclusion bodies as seen from small-scale test expressions. Thus,

refolding was still required. The final protocol used for expression of inclusion bodies was to grow the BL21 (DE3) culture at 37 °C to an $OD_{600} = 0.6$ where 1 mM IPTG was added to induce recombinant protein expression. The culture was allowed to express overnight.

## Purification of FLRT FnIII domains

From the non-optimized gene constructs it was only possible to express the FnIII domains from FLRT1 and FLRT3, but despite very low yields, a purification procedure could be established. Since the recombinant proteins were localized in inclusion bodies, the bacteria pellets were solubilized in denaturation buffer containing 8 M urea followed by immobilization and refolding on-column using a Ni-NTA column, and by decreasing the concentration of urea to 0 M. After washing, proteins were eluted and further purified using SEC (see Fig. 1). From a 1 L culture 1–2 mg protein was finally obtained for further characterization. The purities of the recombinant proteins were validated by SDS-PAGE, which demonstrated that the proteins were ∼99% pure protein (see Fig. 2). Initially, refolding was performed without DTT. However, all three FLRT FnIII domains contain two cysteines. Thus, DTT was included in the denaturation buffer. The addition of DTT had a significant effect on the monomer/dimer ratio as determined by SEC (see Fig. 1, (green curves)).

## Characterization of FLRT FnIII domains
### Mass spectrometry

For all three FnIII domains, MALDI-TOF-MS analysis was performed to ensure correct recombinant protein expression as well as pure sample (see Table 1, Table S1 and Fig. 3). Methionine (Met) has previously been shown to be artificially oxidized due to sample handling both for MALDI-TOF and ESI measurements (*Chen & Cook, 2007*; *Thirumangalathu et al., 2007*), thus while analyzing theoretical and experimental masses, oxidation of one or more Met residues was allowed.

For FLRT3-FnIII MALDI-TOF we observed 1 peak (12232.2 Da) corresponding to the loss of the N-terminal Met. It has previously been shown that, when a small amino acid like glycine is following the N-terminal Met, Met is likely to be cleaved off (*Frottin et al., 2006*). For FLRT2-FnIII 3 peaks were observed, 1 at 11746.3 Da corresponding to the N-terminal degraded by 6 residues (1-MGPISE-6) see Table S1 , and the other 2 at 11789.5 Da and 11827.7 Da most likely corresponding to either carbamylation (+43 Da) from the urea or acetylation (+42 Da), and the same post-translational variant with a bound potassium ion (+38 Da) see Fig. 3. In the case of FLRT1-FnIII 5 peaks were observed at 11122.8 Da, 11254.2 Da, 11947.6 Da, 12077.3 Da and 12106.0 Da (see Fig. 3). Here we observe 2 N-terminally degraded variants (11122.8 Da and 11254.2 Da), an N-terminal Methionine cleaved variant (11947.6 Da), a full length FLRT1-FnIII (12077.3 Da) see Table S1 , and a mass of +28 Da from the full length (not identified here). In conclusion these results demonstrate that the recombinant proteins are correctly expressed, but that they are prone to N-terminal degradation. Notably, the proteins were not stored with protease inhibitors. Addition of protein inhibitors would possibly reduce the degree of the observed N-terminal degradation.

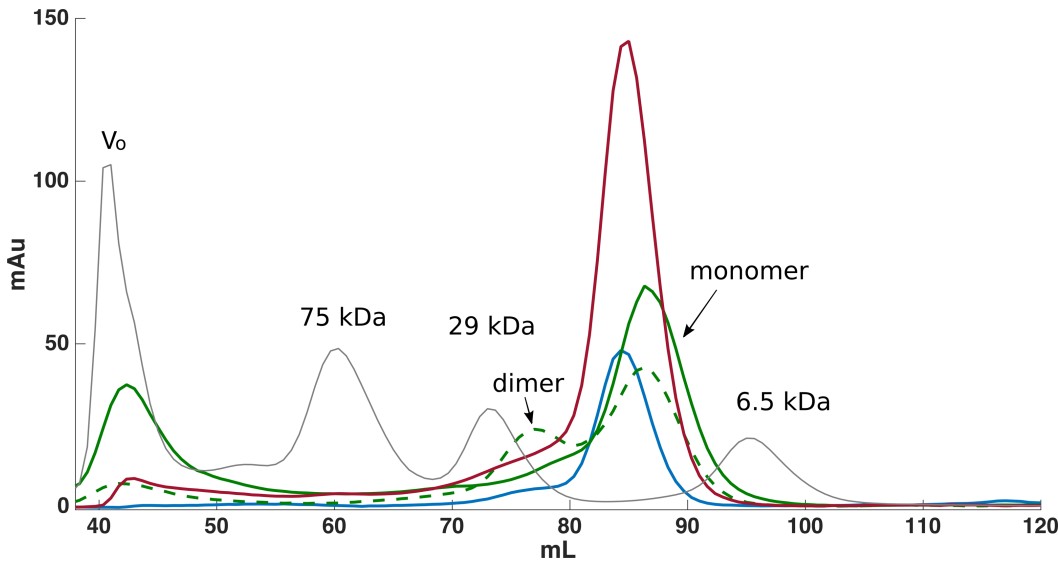

**Figure 1** **Size exclusion chromatography was used for further purification of FLRT-FnIII samples.**
Initially purification was attempted without DTT, which resulted in monomeric and dimeric forms of
FLRT2-FnIII as seen from the SEC curve (green, dash line). After including DTT, the presence of dimeric
forms is significantly reduced although a small fraction is still present (green). Subsequent purifications
of FLRT1- (blue) and FLRT3-FnIII (red) included DTT, and resulted in similar profile with monomers as
the predominant form. Standards in grey (Blue Dextran to determine Void volume, Vo; Conalbumine, 75
kDa; Carbonic Anhydrase, 29 kDa; Aprotenin, 6.5 kDa) were used to calibrate the column, and estimate
the molecular weights (Mw) of the FnIII domains.

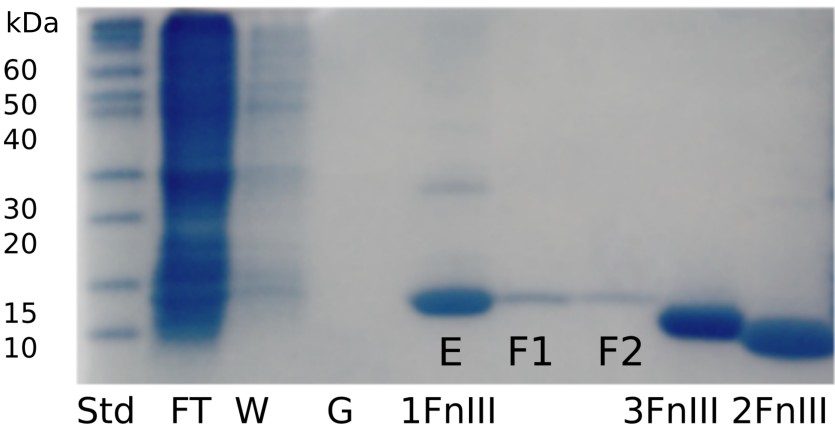

**Figure 2** **SDS-PAGE gel following purification and refolding of FLRT1-FnIII together with purified
and concentrated FLRT2-FnIII and FLRT3-FnIII domains.** To verify purity and to follow expression, re-
folding and purification steps, samples were taken at different steps; flow through the nickel-column (FT),
wash of immobilized sample (W), application of refolding gradient (G), and elution (E). Also, the SEC pu-
rification samples were taken from different fractions (F1 and F2). FLRT2-FnIII and FLRT3-FnIII were
purified and refolded following the same protocol. Concentrated samples can be seen next to FLRT1-FnIII
purification, showing pure proteins. Markers are also shown (Std).

**Table 1  Masses determined from MALDI-TOF.** Theoretical masses are calculated from sequences given in 'Material and Methods' including N-terminal Methionine and C-terminal His$_6$-tag.

| | MALDI-TOF | | | |
| --- | --- | --- | --- | --- |
| | Peak (charge) | Peak (charge) | Peak (charge) | Peak (charge) |
| FLRT1 | 12077.3 | 11947.6 | 11254.2 | 11122.8 |
| Theoretical mass (theor.) | 12073.72 | 12073.72 | 12073.72 | 12073.72 |
| Deviation from theor. | 3.58 | 126.12 | 819.52 | 950.92 |
| FLRT2 | 11827.7 | 11789.5 | 11746.3 | |
| Theoretical mass (theor.) | 12335.15 | 12335.15 | 12335.15 | |
| Deviation from theor. | 507.45 | 545.65 | 588.85 | |
| FLRT3 | 12230.8 | | | |
| Theoretical mass (theor.) | 12356.27 | | | |
| Deviation from theor. | 125.47 | | | |

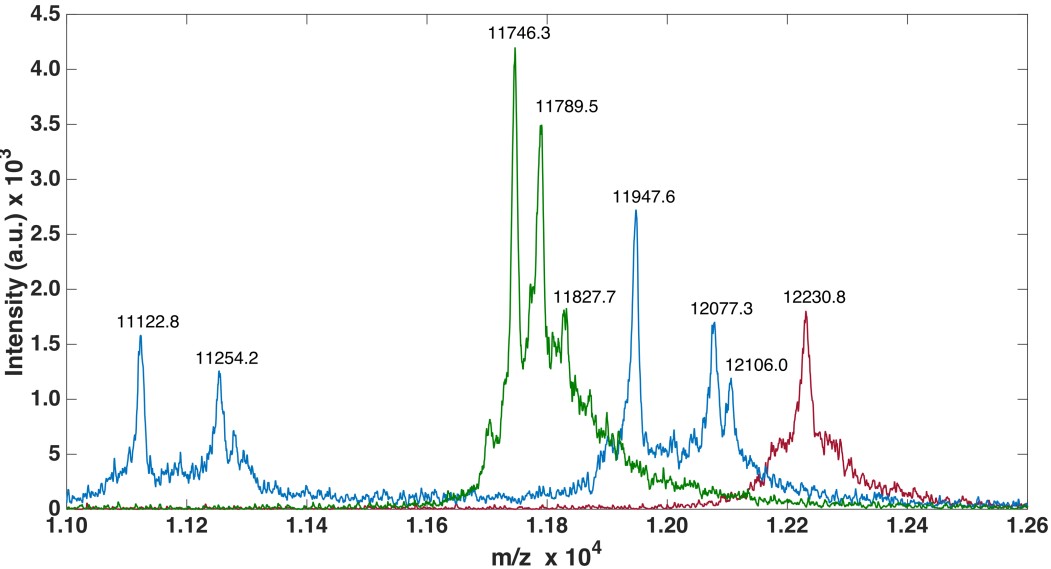

**Figure 3  MALDI-TOF analysis of FLRTs FnIII domains.** MALDI-TOF was performed to validate purity. FnIII domains from FLRTs were all prone to degradation. For FLRT1 (blue) and FLRT2 (green) domains the degradation was more pronounced than for FLRT3 (red).

### Structural characterization by Circular Dichroism and NMR Spectroscopy

Far-ultraviolet CD spectra measurements were performed on all three FnIII domains to confirm that the refolding had been successful (Fig. 4). The spectra of the FLRT1 FnIII and FLRT3 FnIII domains were almost identical with deviations only at wavelengths below 208 nm. The spectra are typical of a classical β-sheet signal with a right twist of the sheet influencing the spectrum from a (positive) maximum at ∼230 nm to a (negative) minimum around 215 nm (*Micsonai et al., 2015*). The maximum ∼230 nm in the profile may have contributions from the aromatic side chains as discussed in earlier studies of other FnIII domains (*Stevens et al., 1987*; *Brumfeld & Werber, 1993*). The spectra thus clearly show that the two domains are folded. Furthermore a 1D proton NMR spectrum, $^1$H-NMR,

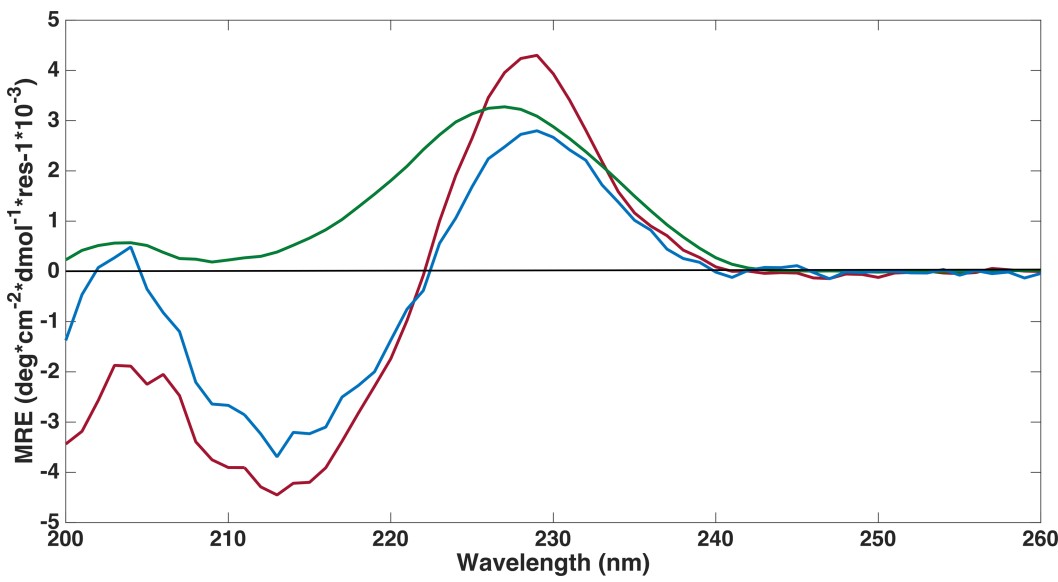

**Figure 4** **Far-UV circular dichroism spectroscopy was performed to assess if proteins were folded.** CD spectroscopy experiments were conducted on refolded FLRT-FnIII domains (FLRT1-FnIII (blue), FLRT2-FnIII (green) and FLRT3-FnIII (red)) in PBS buffer. All samples have a positive signal around ∼230 nm, which is characteristic for fibronectin domains. MRE is mean residue ellipticity.

of FLRT1-FnIII (see Fig. S1 ) resulted in broad dispersion of peaks with two peaks at 10.2 ppm and 10.0 ppm arising from the two tryptophan Nε protons. The two different chemical shifts clearly show that these tryptophans have different local environment and also support the interpretation that the produced FLRT1-FnIII domain is folded. The CD spectrum of FLRT2-FnIII deviates from the spectra of the other two proteins. However, it also has a maximum at ∼230 nm indicating that the domain is folded. CD signals from antiparallel β-sheet are relatively weak compared to an α-helical signal. The CD profile of FLRT2 FnIII still has an antiparallel β-sheet signal, but in contrast to the FLRT1 and FLRT3 FnIII domains, the FLRT2 FnIII domain is estimated to contain a larger fraction of the antiparallel β-sheet (47.6%). Of this fraction 30.2% is estimated to be anti3 (right-twisted) β-sheet, which is 6–7% more than for the FLRT1 and FLRT3 FnIII domains (see Table 2). This could explain the signal differences between the CD profiles from the FnIII domains. Due to the differences in the CD spectra, it cannot be excluded that a larger part of the FnIII domain from FLRT2 is folded differently compared to the FnIII domains from FLRT1 and FLRT3, or that the protein is only partially folded.

## Stability of the FnIII domains
Due to the differences in CD spectra, it could not be excluded that FLRT2 FnIII might only be partially folded. In order to investigate this possibility, thermal unfolding of all three domains was performed by variable temperature measurement at a fixed wavelength (228.5 nm) where a strong CD signal was observed (see Figs. 4 and Fig. 5 for the full CD spectra). Both FLRT1 FnIII and FLRT3 FnIII unfolded at ∼68−69 °C. In contrast, FLRT2 FnIII unfolded at ∼42 °C. To explain this significant difference, we hypothesized

**Table 2  Estimated secondary structure content.** The BESTSEL server (*Micsonai et al., 2015*) is developed to characterize secondary structure content from CD spectra. It is especially developed for analysis of antiparallel $\beta$-sheet folded proteins. Thus, it provides estimates of the percentages of left-(Anti1) and right-twisted (Anti3) as well relaxed (Anti2) antiparallel $\beta$-sheets.

| | Helix | Antiparallel | Parallel | Turn | Others |
|---|---|---|---|---|---|
| FLRT1FnIII | 1.9 | 43.3 | 0.0 | 9.0 | 45.8 |
| FLRT2FnIII | 0.0 | 47.6 | 0.0 | 8.4 | 43.9 |
| FLRT3FnIII | 0.0 | 41.5 | 0.0 | 6.5 | 51.9 |

| | | | Degree twisted | | |
|---|---|---|---|---|---|
| | Helix1 | Helix2 | Anti1 (left-twisted) | Anti2 (relaxed) | Anti3 (right-twisted) |
| FLRT1FnIII | 0.7 | 1.2 | 0.8 | 19.5 | 23.0 |
| FLRT2FnIII | 0.0 | 0.0 | 0.0 | 17.4 | 30.2 |
| FLRT3FnIII | 0.0 | 0.0 | 0.0 | 17.7 | 23.9 |

that a Cys–Cys bridge from strand F to strand G is stabilizing the FnIII domain of FLRT1 and FLRT3, but not FLRT2. To test this hypothesis, identical variable temperature CD measurements were recorded with the only difference that 1 mM DTT had been added to the samples. The addition of DTT had a significant impact on the unfolding of FLRT1 FnIII and FLRT3 FnIII. In the presence of DTT the FLRT1 and FLRT3 FnIII domains unfolded at ~48 and 38 °C, respectively. They were destabilized to a degree where a plateau at low temperature (20 °C) was not established, making a proper $T_m$ value difficult to assess. In the case of FLRT1 FnIII domain the signal/noise was relatively low, but it is clear that in the presence of DTT the domain was less stable. In contrast, the addition of DTT did not affect the unfolding temperature of FLRT2 FnIII domain (see Fig. 5C). The domain still unfolds with $T_m$ ~42 °C, indicating the absence of disulphide formation although two cysteines are present on F and G strands as for the other two domains (see Fig. 5C). A sequence alignment of the three FnIII domains shows that the FLRT2 FnIII domain lacks a residue between the Cys residues, when compared to FnIII domains of FLRT1 and FLRT3 (see Fig. S2). In the case of FLRT2, it could be a distance restriction that prevents the formation of a cysteine bridge. Another but less likely situation would be that the disulphide bridge is formed, but does not contribute further to the stability of the FLRT2 FnIII domain. The order of stability of FnIII domains in the presence of DTT is FLRT3 < FLRT2 < FLRT1. The differences, especially between the FnIII domains from FLRT1 and FLRT3, which are quite significant, could be due to the differences between the hydrophobic cores of the domains. From the sequence alignment of the domains, it can be hypothesized that the extra aromatic residues in the FLRT1 FnIII domain stabilizes the core more when compared to the FLRT3 FnIII domain. A final explanation for the results obtained is that the FLRT2 FnIII domain is incorrectly folded, so that the native Cys–Cys bridge cannot be formed, giving a rationale for the differences in CD-spectra compared to the two other FnIII domains. From mass spectrometry it was possible to observe N-terminal terminal cleavage of FLRT2 FnIII, which might affect the folding, or be a consequence of partial

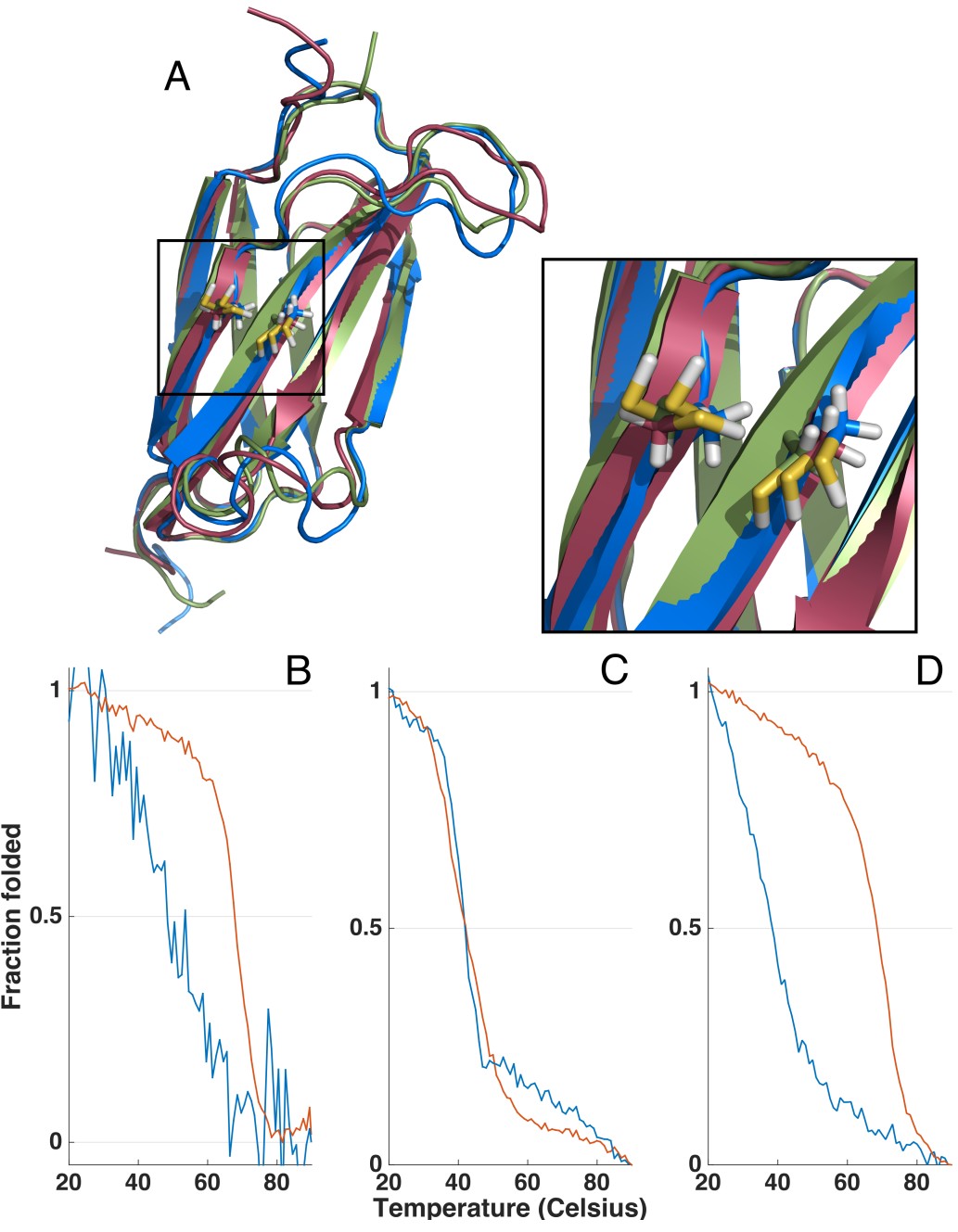

**Figure 5** **Homology models and thermal unfolding of the FnIII domains from the FLRT family as monitored by the CD signal at 228.5 nm, normalized to vary between 1 and 0 for the initial and final data point in each series.** The I-TASSER server (*Yang et al., 2014*) was used to predict the topology and the location of the two cysteins located in the FnIII domains of FLRT1 (blue), FLRT2 (green) and FLRT3 (red) (A). In all the I-TASSER modelled FLRT-FnIII domains the two cysteines are located in closed vicinity on each $\beta$ stand (F and G) (A, zoom in). For FLRT1-FnIII (B) and FLRT3-FnIII (D) there is a large difference in thermal unfolding curves in the presence (blue line) and absence (red line) of reducing agent, DTT. In contrast, thermal unfolding of FLRT2-FnIII is not affected by DTT (C) as for FLRT1-FnIII and FLTR3-FnIII.

unfolding. In each case, with or without DTT, the measurement was carried out from 90 to 20 °C to determine the folding reversibility. However, none of the domains exhibited any reversibility and thus remained unfolded after the first thermal unfolding.

## Template-based protein structure prediction of FLRT FnIII domains

With few exceptions (*Huber et al., 1994*), FnIII domains of known structures do not contain Cys–Cys bridges, while the FLRT domains investigated here do. The Cys residues are at position 75 and 93 in the sequences of FLRT1 and FLRT3 FnIII, and position 75 and 92 in FLRT2 FnIII. Since the stability investigations indicate a possible involvement of disulphide bridges in the stability, we wanted to investigate if the spatial arrangement allows for Cys–Cys bridging by performing template-based protein structure prediction using I-TASSER. A detailed description of I-TASSER is given elsewhere (*Zhang, 2007*; *Wu, Skolnick & Zhang, 2007*; *Yang et al., 2014*). Briefly, I-TASSER predicts protein structures from primary sequences, first by identifying structural templates from the PDB library with LOMETS, next by performing an iterative Monte Carlo simulation and model refinement to construct the topology of the protein. Up to five models are constructed and validated by confidence score (*C*-score). *C*-scores should be in the range between −5 and 2, with more positive values being the better and *C*-scores over −1.5 indicating usually a correct fold. The best models all had *C*-scores better than −0.5 see Table S2. For the FLRT2 FnIII domain we obtained four models, which is a good sign of convergence, and thus reliability of the model. I-TASSER also returns a TM score, which is a sequence length-independent metric for measuring structure similarity in the same SCOP/CATH fold family. The TM values were in the range between 0 and 1 (a TM-score > 0.5 indicating a model with correct topology and a TM-score < 0.17 indicating random similarity). For all three FnIII domains the models scored TM values > 0.66 (see Table S2). The best I-TASSER model is used in a search against structure in the PDB library to identify structural homologues. This search is different compared to the former, which is a sequence-based threading search while the latter is a structural search.

Several structures of FnIII domains have been solved within the last decades, and despite high tertiary structure similarities they tend to share low sequence identity. This is also evident from the I-TASSER structure prediction where sequence identities with the used templates range from 12.6 to 23.2%. All top solutions had sequence coverage from 91.3 to 94.1%. The tertiary structure of the FnIII domains consist of a β-sheet built up by B, C, E and F β-strands, and a β-sheet composed of A, C' and G β-strands. A hydrophobic core stabilizes the domain. The topology of the FnIII domain has previously been shown to have conserved Trp residues as well as a hydrophobic pattern building up the hydrophobic core. When inspecting the sequences of the FLRTs FnIII domains it can be seen the also contain this hydrophobic pattern (see Fig. S3). Taken together, the modelling of the FLRT FnIIIs seems reliable, within the limits imposed by low sequence identity with the used templates. All models modelled by I-TASSER placed the two Cys residues in close vicinity, even though the templates used for modeling did not contain cysteines. This suggests that a disulphide bridge could be formed between the F and G strand (see Fig. 5), explaining the effect of DTT in the stability of the FLRT1 and FLRT3 FnIII domains. For the FLRT2

FnIII domain the situation is not as clear. The thermal unfolding showed no effect of DTT, and this could be due to partial unfolding or misfolding, but a disulphide bond may be formed *in vivo*.

### Interactions between with FGFR1 and FLRT FnIII domains

Since the *Xenopus* FLRT3 FnIII domain has been shown *in vivo* by truncation experiments to interact with FGFR1 (*Böttcher et al., 2004*), we tried to see if interaction between human FLRT FnIII domains and FGFR1 could be demonstrated *in vitro*. Surface plasmon resonance (SPR) analysis with FGFR1β immobilized on the chip, and FLRT FnIII domains in solution demonstrated the expected binding between FGFR1β and FLRT FnIII domains (see Fig. S4). On the same chip an Ig domain had been immobilized as negative control to detect unspecific binding. No unspecific binding was observed for any of the tested FnIII domains (see Fig. S4). By steady state fitting we roughly estimated the $K_d$'s for the interaction between FGFR1β and the FnIII domains. The $K_d$s are all in the μM range (see Fig. S4), which corresponds to $K_d$ values obtained for other FGFR-FnIII domains interactions (*Christensen et al., 2006*). The $K_d$s should only be taken as indicative as the measurements were only performed twice. Interestingly, although the FLRT2 FnIII domain shows lower affinity than the other two, the $K_d$ is in a similar range suggesting that the domain is at least partly correctly folded.

## CONCLUSION

Here we present a protocol for expression and on-column refolding of FLRT FnIII domains. It can be concluded that codon optimization had significant effects on the expression levels, when compared to non-codon optimized constructs. Moreover, it can be concluded that on-column refolding of FLRTs-FnIII domains can be performed successfully as seen from CD experiments and NMR studies, but also that protease inhibitors should be included in future studies, since the domains are prone to cleavage. Also, we concluded from a combination of modelling and stability studies that a Cys–Cys bridge between the F and G strands might be necessary to stabilize the FnIII domains. Finally, we show that all three recombinant human FLRT FnIII domains *in vitro* bind FGFR1b with $K_d$ values in the μM range. Binding of FLRT proteins to FGFRs have been reported to include both extracellular and intracellular interactions (*Wei et al., 2011*), but our data demonstrate that FLRT FnIII domains alone are sufficient to mediate FLRT-FGFR interaction *in vitro*. We speculate that differences in the primary, secondary or tertiary structures of FLRT FnIII domains might contribute to different binding affinities between different members of FLRT and FGFR families, hence ensuring variability in the biological responses to FLRT-FGFR interactions.

## ACKNOWLEDGEMENTS

We will like to thanks Christian Tortzen for assistance using the NMR spectrometer, and laboratory technicians Annette Andersen and Uriwan Ngamrabiab Adamsen, for assistance using mass spectrometry. Also we like to thanks Novo Nordisk contact Frederik Öberg for discussions.

### Funding

This work was supported by the Lundbeck Foundation (Grant: R165-2013-15975 to KKR), the University of Copenhagen through the CoNeXT project, "Læge Sofus Carl Emil Friis og Hustru Olga Dorus Friis' Legat". A PhD studentship to MHF was funded by the University of Copenhagen and the Novo Scholarship program for funded LY. The funders had no role in study design, data collection and analysis, decision to publish, or preparation of the manuscript.

### Grant Disclosures

The following grant information was disclosed by the authors:
Lundbeck Foundation: R165-2013-15975.
University of Copenhagen.
Læge Sofus Carl Emil Friis og Hustru Olga Dorus Friis' Legat.
Novo Scholarship program.

### Competing Interests

The authors declare there are no competing interests.

### Author Contributions

- Lila Yang and Maria Hansen Falkesgaard performed the experiments, analyzed the data, wrote the paper, prepared figures and/or tables.
- Peter Waaben Thulstrup performed the experiments, analyzed the data, contributed reagents/materials/analysis tools, reviewed drafts of the paper.
- Peter Schledermann Walmod and Leila Lo Leggio conceived and designed the experiments, contributed reagents/materials/analysis tools, reviewed drafts of the paper.
- Kim Krighaar Rasmussen conceived and designed the experiments, performed the experiments, analyzed the data, contributed reagents/materials/analysis tools, wrote the paper, prepared figures and/or tables, reviewed drafts of the paper.

### Data Availability

   The raw data has been supplied as Supplemental Information 1.

### Supplemental Information

Supplemental information for this article can be found online at http://dx.doi.org/10.7717/peerj.3550#supplemental-information.

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
