# Peer review of "Expression, refolding and spectroscopic characterization of fibronectin type III (FnIII)-homology domains derived from human fibronectin leucine rich transmembrane protein (FLRT)-1, -2, and -3"

_PeerJ, doi:10.7717/peerj.3550_

## Round 0.1 · original submission · Major Revisions

Although reviewer #1 was very positive, reviewers #2 and #3 raised several critical issues that need to be addressed.

Reviewer 1 ·

Basic reporting

Please check the manuscript for grammatical errors/typographical errors.

Experimental design

Mention whether Figure 2 was generated using Coomassie staining or another stain?

Please state clearly (in the conclusion section) how the research presented in the manuscript contributes to the understanding of FLRT biology.

Validity of the findings

no comment

Additional comments

Overall, the experiments in the manuscript are well-thought out and are presented in a logical order.
The authors present multiple scenarios to explain the variations seen in the CD spectra obtained from the FnIII domains for FLRT2 compared to FLRT1 and 3.
The authors also mention experiments/experimental conditions that gave negative results, which is particularly relevant if these experiments need to be replicated by another research group.
However, the relevance of the paper will be enhanced if the authors add a few lines explaining the biological significance of their work

Reviewer 2 ·

Basic reporting

The organization of the data is fine; however, the writing is poor. There are too many typos, grammar and other small issues to fix. Figures are not labeled properly. I suggest that the authors find a professional scientific writer or native English speaker to carefully go through the manuscript.

Experimental design

The aim and scope is good, the subject of the research FnIII domains have important functions in biology. The description in methods has flaws, mainly with how the authors used thermal unfolding to determine Tm.

Validity of the findings

The data will be good for the general public.

Additional comments

Yang et al. presented in this manuscript a protocol for purification and subsequent biophysical characterization of three FLRT-FnIII domains. The authors managed to obtain purified FnIII domains by solubilization of inclusion bodies, Ni-NTA affinity chromatography and SEC. To validate whether the obtained proteins are folded, the authors used mass spectrometry, CD and 1H-NMR spectroscopy. The authors also explored the stability of the obtained proteins by thermal denaturation monitored by CD signal. The binding of FnIII domains to FGFR1 was also verified by SPR. The organization of the data is fine; however, the writing is poor. There are too many typos, grammar and other small issues to fix. I suggest that the authors find a professional writer or native English speaker to carefully go through the manuscript.

Aside from the small things, there are several concerns on the interpretation of the data in this manuscript.

1. In line 39, the authors describe FLRTs as proteins located on the surface of neurons. That is just oversimplification of the situation as FLRTs are expressed elsewhere.
2. The second problem is with mass spec experiments (table 1), that the authors used MALDI-TOF MS to characterize FLRT1 but used ESI-MS to characterize FLRT2 and FLRT3. The authors didn’t explain the logics of using different techniques to characterize different proteins. I suggest the authors to stick to ESI-MS for all 3 purified proteins or obtain both MALDI-TOF MS and ESI-MS data for all 3 proteins.
3. In main text line 255, the interpretation of the observed mass for FLRT1 is strange. The authors suggest that the observed mass come from two truncated forms of FLRT1, but because the protein is purified by Ni-NTA affinity chromatography, there is no reason that the His tag is truncated. Also it is unlikely that the truncation happened during purification, otherwise simply adding some protease inhibitor would solve the problem. If part of the protein got lost because of MALDI-TOF, the authors could simply switch to ESI-MS for characterization. The authors need to find a better answer to explain the observed mass for FLRT1.
4. In Figure 3 CD spectra, the authors need to add the unit for the CD signal. The correct unit should be like (deg cm2 dmol-1 res-1 x 10-3).
5. The quality of the CD spectra in Figure 3 was not good for <210 nm. Instead of PBS, a phosphate buffer without Cl- is a better choice for CD experiments.
6. In main text line 291, also Figure 4, the method for the determination of Tm for the 3 purified proteins is not described. Based on the description in Figure 4B-4D (the y axis was not labeled) I believe the authors took the lowest point in CD as the 100% unfolded and the highest point in CD as 100% folded. However, for the two state transition both the folded state and unfolded state have their baselines, and often times these baselines are not flat. For example in figure 4C, the protein is basically 100% unfolded at 60C, as well as at 80 C, and the CD signal for these two temperatures are not the same, but they fall on the same baseline. The authors should explain further on how they determine the Tm of a protein in thermal unfolding experiments and took the baselines into consideration.
7. In main text line 293, the authors monitored both 228.5 and 213 nm for thermal unfolding experiments, but it is unclear which wavelength they used for producing Figure 4B-4D.

Annotated reviews are not available for download in order to protect the identity of reviewers who chose to remain anonymous.

Reviewer 3 ·

Basic reporting

Overall a well structured report with a straightforward aim of expressing FN3 domains of FLRT for future biophysical studies. The manuscript is let down by (1) sloppy model building and (2) many instances of poor grammar and sentence construction, which often obscures the meaning, and would benefit greatly from editing. I have noted some examples below:

Line 89 “number of diseases” (plural)
line 92-93 “among which are FLRTs”?
line 291 should read “…..only be partially folded”
line 301 “They were even so” delete “even”
line 303 should be re-phrased, currently it does not make sense “but the tendency of a destabilized folding is evident.”
line 332 “However, an atypical thing with these FnIII domains in contrast to other known human FnIII domains is the presence of cysteines, which are located on the F and G strands (see Fig. S2).”. Needs rewriting.
line 311: what does “The order of unfolding in presence of DTT is FLRT3<FLRT2<FLRT1.” mean?

Line 295-297: “To explain these significant differences we hypothesize that a Cys-Cys bridge from strand F to strand G is stabilizing the fold in FLRT1 and FLRT3, but not FLRT2.” What is the reasoning behind this?

Experimental design

The section on homology modelling is written poorly, one does not get a feel for how reliable the models are. Although the authors quote a PHYRE confidence level of 99% (based on coverage), the sequence identity with the templates ranges from 9 to 25%, and for FLRT2 the top model has only a 9% sequence ID with the template. These values are low and considered challenging for homology model building. The authors refer to secondary structure but neither sequences nor model structure are annotated with these.
Can the postulated disulphide bond be accurately modelled? e.g. is shown in fig 4A but how was this constructed?
This part of the study severely compromises the work, and should be rewritten entirely. I am skeptical of the modelling and how it should be interpreted, so at the very least more details should be shown here to convince the reader of the quality and reliability of the models, especially as the authors try to attempt to explain the thermal melting data, and also the role of a putative disulphide bridge in the domain stability.

Validity of the findings

See comments above - validity of experimental results is overall sound. Homology modelling section is poor.

---

## Round 0.2 · accepted · Accept

Thank you very much for addressing all critical points raised by all reviewers.